# Transcriptional Regulators Controlling Virulence in *Pseudomonas aeruginosa*

**DOI:** 10.3390/ijms241511895

**Published:** 2023-07-25

**Authors:** Ana Sánchez-Jiménez, María A. Llamas, Francisco Javier Marcos-Torres

**Affiliations:** Department of Biotechnology and Environmental Protection, Estación Experimental del Zaidín-Consejo Superior de Investigaciones Científicas, 18008 Granada, Spain; ana.sanchez@eez.csic.es

**Keywords:** *Pseudomonas aeruginosa*, pathogenesis, virulence, transcription regulators, sigma factor, one-component system, two-component system, quorum sensing

## Abstract

*Pseudomonas aeruginosa* is a pathogen capable of colonizing virtually every human tissue. The host colonization competence and versatility of this pathogen are powered by a wide array of virulence factors necessary in different steps of the infection process. This includes factors involved in bacterial motility and attachment, biofilm formation, the production and secretion of extracellular invasive enzymes and exotoxins, the production of toxic secondary metabolites, and the acquisition of iron. Expression of these virulence factors during infection is tightly regulated, which allows their production only when they are needed. This process optimizes host colonization and virulence. In this work, we review the intricate network of transcriptional regulators that control the expression of virulence factors in *P. aeruginosa*, including one- and two-component systems and σ factors. Because inhibition of virulence holds promise as a target for new antimicrobials, blocking the regulators that trigger the production of virulence determinants in *P. aeruginosa* is a promising strategy to fight this clinically relevant pathogen.

## 1. Introduction

*Pseudomonas aeruginosa* is clinically recognized for being an opportunistic pathogen of humans that causes severe infections, mainly respiratory tract infections but also urinary tract, ear, and eye infections, and bacteraemia and sepsis [1]. As an opportunistic pathogen, *P. aeruginosa* is rarely pathogenic in healthy individuals, and the development of infections usually depends on epithelial injuries or impaired immune defenses [2]. Nevertheless, *P. aeruginosa* is a primary source of hospital-acquired infections especially in mechanically ventilated patients in intensive care unit, and burns, open wounds, and post-surgery patients [3]. Moreover, it is the main reason for lung function deterioration and mortality in cystic fibrosis (CF) patients, who are chronically infected with this pathogen [4]. Infections caused by *P. aeruginosa* are often life-threatening, and treatment can be challenging because this bacterium is intrinsically resistant to multiple antibiotics and can easily acquire new resistances [5]. In fact, antibiotic resistance among *P. aeruginosa* has escalated globally over the past three decades, and several outbreaks in hospitals have highlighted the need for controlling multi-drug resistant *P. aeruginosa* infection and spread [6]. For this reason, the World Health Organization (WHO) declared this bacterium a priority pathogen, demanding research and the development of new strategies for its eradication.

*P. aeruginosa* owes its pathogenic versatility to a large arsenal of cell-associated and secreted virulence factors that enable host colonization and protection from host innate immune defenses. The main virulence factors and hallmark capabilities associated with virulence produced by this pathogen include (1) bacterial motility and attachment factors; (2) biofilm formation factors; (3) extracellular invasive enzymes and secreted toxins; (4) toxic secondary metabolites; and (5) iron acquisition systems and factors controlling iron homeostasis [7] (Table 1).

### 1.1. Motility and Attachment Factors 

*P. aeruginosa* motility depends on the flagellum and the type IV pili. These appendages control the motility of the bacterium (Table 1) and are involved in important virulence processes including searching for a desirable surface and allowing cell-surface contact, transitioning from a planktonic to a sessile lifestyle, and promoting biofilm formation [8]. Flagella are polar protein complexes that use hydrodynamic forces to propel the bacterial cell in liquid (swimming motility) and semi-solid surfaces (swarming motility) [8]. There are two main subfamilies of type IV pili, namely, type IVa and type IVb. Twitching motility on solid surfaces is driven by type IVa pili, and they cooperate with flagella to perform swarming motility. Type IVa pili are composed of a large number of PilA pilin subunits and are retractile thanks to the action of PilT ATPase [9]. Type IVb pili, which are also known as tight adherence (Tad) pili, are composed of Flp pilin subunits. Tad pili are non-retractile, so they are mainly involved in the attachment of *P. aeruginosa* to abiotic and biological surfaces, a critical step for proper colonization and infection of host tissues and for biofilm formation [9,10]. The major adhesins, known as chaperone-usher pathway (CUP) pili, are also important for *P. aeruginosa* attachment to host surfaces. This bacterium produces five different CUP systems (from CupA to CupE) (Table 1) [11]. Other proteins important for *P. aeruginosa* attachment to solid surfaces and adhesion to host cells are the biofilm associated protein BapA and lectins LecA and LecB (Table 1) [12,13]. 

### 1.2. Biofilm Formation Factors 

The ability of *P. aeruginosa* to form a biofilm is key for the establishment of chronic infections. Although it is expected that other species present in the host participate in the formation of biofilms with *P. aeruginosa*, most experimental data on virulence factors and their regulation were obtained using analyses of biofilms from *P. aeruginosa* monocultures. The main constituents of *P. aeruginosa* biofilms are exopolysaccharides (EPSs), i.e., alginate, Pel, and Psl, and extracellular DNA (eDNA) [14] (Table 1). Alginate is considered an important virulence factor of *P. aeruginosa* because overproduction of this EPS results in the clinically relevant mucoid phenotype [15]. Mucoid cells grow as compact microcolonies, which confer protection against antibiotics and immune system attacks, thus contributing to the long-term persistence of *P. aeruginosa* in chronic infections [15]. This phenotype also leads to the establishment of concomitant airway inflammation and is the main cause of mortality in CF patients [15]. Rhamnolipids, which are extracellular secondary metabolites, also participate in different stages of *P. aeruginosa* biofilm development, from the earliest cell-to-surface interactions to the maintenance and dispersion of the biofilm architecture [16] (Table 1). Rhamnolipids are considered important virulence factors because they mediate the active dispersal of cells from biofilms, thus promoting the colonization of new niches, and act in biofilms as protective agents against phagocytes [17]. Moreover, the amphiphilic properties of rhamnolipids make them capable of disrupting host cell tight junctions when being incorporated into host membranes [18]. In addition to rhamnolipids, *P. aeruginosa* produces glycine betaine, another bacterial surfactant that confers cells resistance to osmotic stress, protects them from phagocytosis, and increases the survival of *P. aeruginosa* in the mouse lung during infection [19].

**Table 1 ijms-24-11895-t001:** Transcriptional regulators of the main *P. aeruginosa* virulence hallmarks *.

Virulence Hallmark	Virulence Factors	Genes Involved	Function in Virulence	Main Transcriptional Regulator(s)	References
Bacterial motility and attachment	Flagellum	*flg*, *fli*, and *flh* gene clusters	Swimming and swarming motility	AmrZ, PilS-PilR, FleS-FleR, σ^FliA^	[20,21,22]
	Type IV pili	*flp*, *tad*, *afimU* *fimV*, and *pilABCD*	Twitching motility and attachment to solid surfaces	AmrZ, PchR, PilS-PilR, PprA-PprB, GacS-GacA, BfiS-BfiR, GtrS-GltR, σ^SigX^	[12,22,23,24,25,26,27,28]
	Adhesins	*bapA*, *cupA*, *cupB*, *cupC*, *cupD*, and *cupE* gene clusters	Attachment to host cells and solid surfaces	RocS-RocR-RocA1, RcsC-RcsB, PvrS-PvrR, PprA-PprB, GacS-GacA, BfiS-BfiR, GtrS-GltR, σ^SigX^, σ^RpoS^	[11,23,24,26,27,29,30,31]
	Lectins	*lecA* and *lecB*	Adherence to epithelial cells	PqsR, Fur, PchR, σ^AlgT^	[28,32,33,34]
Biofilm formation	Esopolysaccharide alginate	*alg* gene cluster	Component of the extracellular matrix	AmrZ, PchR, FimS-AlgR, KinB-AlgB, σ^AlgT^, σ^RpoH^, σ^RpoS^, σ^SbrI^	[28,35,36,37,38,39,40]
	Exopolysaccharides Pel and Psl	*pel* and *psl* gene clusters	Component of the extracellular matrix	GacS-GacA, BfiS-BfiR, GtrS-GltR, σ^AlgT^, σ^RpoS^	[24,26,27,33,36,37,41]
	eDNA	*phdA*	Component of the extracellular matrix	BfmS-BfmR	[42]
	Rhamnolipids	*rhlAB* and *rhlC*	Biosurfactant	RhlR, PtxR	[43,44]
	Glycine betaine	*betA* and *betB*	Biosurfactant	BetI, GbdR	[45]
Extracellular invasive enzymes and toxins	Elastases	*lasA* and *lasB*	Degradation of host elastin and collagen	LasR, RhlR, PtxR, PchR, FleS-FleR, σ^AlgT^	[28,43,44,46,47,48,49]
	Protease IV	*piv/prpL*	Degradation of host fibrinogen	Fur, σ^PvdS^, σ^FpvI^	[50,51]
	Alkaline protease	*aprA*	Degradation of host transferrin and matrix-associated proteins	LasR, PvrA, σ^SigX^, σ^AlgT^	[23,46,47,52]
	Phospholipases	*plcH*, *exoU*, *pldA*, *pldB*, and *tplE*	Degradation of host cell membranes and lung surfactants	ExsA, GbdR, PvrA, SphR, GacS-GacA BfiS-BfiR, GtrS-GltR, σ^FliA^	[45,52,53,54]
	Exotoxin S and T	*exoS* and *exoT*	Cytotoxicity and cytoskeleton disruption	ExsA, GacS-GacA, BfiS-BfiR, GtrS-GltR, σ^FliA^	[24,26,27,53,55]
	ExoY adenylate cyclase	*exoY*	Disruption of actin cytoskeleton and increased host cell permeability	ExsA, GacS-GacA, BfiS-BfiR, GtrS-GltR, σ^FliA^	[24,26,27,53,55]
	Exotoxin A	*exoA/toxA*	Inhibition of host protein synthesis	PtxR, PtxS, GtrS-GltR, FleS-FleR, σ^PvdS^	[48,51,56,57]
	Pro-inflammatory toxins	*lptA*, *lptE*, and *osmE*	Induction of host inflammatory response	σ^AlgT^	[58]
Toxic secondary metabolites	Hydrogen cyanide	*hcnABC*	Arrest respiration in host cells	LasR, RhlR, PchR, AmpR	[28,47,59]
	Phenazine and pyocyanin	*phzABCDEFG* *phzM*	Cause oxidative stress and cytotoxicity	RhlR, PqsR, CdpR, PvrA, PchR, AmpR	[28,43,44,52,59,60]
Acquisition and homeostasis of iron	Pyoverdine	*pvd* gene cluster	Siderophore production and iron acquisition	Fur, AmpR, σ^PvdS^, σ^FpvI^	[59,61,62]
	Pyochelin	*pchDCBA*, *pchEF*, and *fptABCX* gene clusters	Siderophore production, iron acquisition, and tissue damage	Fur, PchR	[61]
	Heme and hemophore	*phu*, *has*, and *hxu* gene clusters	Iron acquisition from heme	Fur, AmpR, σ^HasI^, σ^HxuI^	[59,61,62]
	Iron storage and detoxification	*prrF1*, *prrF2*, *brfB*, *sodB*, and *katA* genes	Prevent iron accumulation and toxicity	Fur, PchR	[28,61]

* Adapted from [7].

### 1.3. Extracellular Invasive Enzymes and Toxins

Extracellular invasive enzymes that degrade a variety of host connective tissues and immune components (Table 1) are essential in the first steps of host tissue colonization by *P. aeruginosa*. Particularly important to the pathogenicity of this bacterium are extracellular proteases, including elastase B (LasB, pseudolysin), elastase A (LasA, staphylolysin), protease IV (PIV, also known as lysyl endopeptidase PrpL), and alkaline protease (AprA, aeruginolysin) (reviewed in [63]). LasB has elastinolytic activity toward human elastin and also degrades collagens and several components of innate and adaptive immune defenses like tumor necrosis factor-α (TNF-α), interferon-γ (IFN-γ), interleukin-2 (IL-2), and the surfactant proteins A and D (SP-A and SP-D) required for phagocytosis of pathogens. Phagocytic evasion is also enhanced by the alkaline protease AprA, which inhibits chemotaxis of neutrophils and degrades complement proteins and cytokines (i.e., IFN-α, TNF-γ, and IL-6). The PIV/PrpL protease cleaves fibrinogen, thus playing a role in tissue damage and invasion by *P. aeruginosa*. This protease can also degrade plasminogen, immunoglobulin, complement proteins, and host antimicrobial peptides. The LasA, LasB, and PIV/PrpL proteases are secreted to the extracellular medium by the *P. aeruginosa* Xcp type II secretion system (Xcp-T2SS), while AprA is secreted by the type I secretion system (T1SS) [63].

Other important extracellular invasive enzymes produced by *P. aeruginosa* are phospholipases, i.e., lipolytic esterases used by the pathogen to hydrolyze eukaryotic phospholipids (among other functions) (Table 1). This includes the Xcp-T2SS-secreted phospholipase C PlcH, which can hydrolyze several different phospholipids with preference for choline containing phospholipids (i.e., sphingomyelin and phosphatidylcholine) [64]. Hydrolysis of sphingomyelin and phosphatidylcholine yields phosphocholine and ceramide and diacylglycerol, respectively, which are involved in important signal transduction cascades in mammalian cells (i.e., growth, differentiation, apoptosis, proliferation, inflammation). Moreover, PlcH has hemolytic activity and causes hemolysis of human and erythrocytes. This activity is enhanced by the ceramidase CerN encoded within the *plcH* gene cluster [65]. Thus, PlcH promotes *P. aeruginosa* dispersion during infection by providing host-derived nutrients (i.e., choline and iron from erythrocytes) and by triggering signaling pathways that lead to inflammation. Importantly, PlcH is highly lethal to endothelial cells and inhibits angiogenesis (i.e., formation of new blood vessels) [66]. 

A major pathogenicity strategy of *P. aeruginosa* is the production of enzymes and toxins that induce cytotoxicity and various forms of cell death in target host cells [67]. ExoU (for extotoxin U) is a potent cytotoxic A2 phospholipase capable of causing rapid cell death in eukaryotic cells by producing a loss in plasma membrane integrity and necrosis [68] (Table 1). The production of ExoU was found to correlate with acute lung injury and sepsis in an animal model of pneumonia and in immunocompromised patients [69,70]. ExoU killing seems to be directed against phagocytes and epithelial barriers, thus promoting bacterial persistence and dissemination. ExoU is secreted via the type III secretion system (T3SS), a needle complex also designated as the injectosome used by *P. aeruginosa* to inject toxic effectors directly into the cytosol of host cells [68]. Other T3SS-secreted toxins include ExoS, ExoT, and ExoY [68] (Table 1). ExoS and ExoT are bifunctional proteins consisting of an N-terminal GTPase-activating protein (GAP) domain and a C-terminal ADP-ribosyltransferase (ADPRT) domain. The GAP domain in both toxins targets small GTPases required by eukaryotic cells to coordinate and maintain the actin cytoskeleton [68]. The ADPRT domain of ExoS targets many host proteins, thereby producing a number of adverse effects on host cells (i.e., actin cytoskeletal disruption associated with cell rounding, inhibition of DNA synthesis, vesicular trafficking and endocytosis, and cell death) [68]. The ExoT ADPRT domain targets a distinct and more limited number of host proteins, and its activity results in the inhibition of cell migration, adhesion, and proliferation, thus blocking phagocytosis and disrupting epithelial barriers that facilitate bacterial dissemination. Although the functions of ExoS have been thought for a long time to be only cytotoxic and phagocytic, it was recently shown that this toxin also promotes the internalization of *P. aeruginosa* into eukaryotic cells and protection of the intracellular niche [71]. The last T3SS-secreted toxin with a known role during infection and host colonization is ExoY, an adenylate cyclase that increases the intracellular cAMP concentration in eukaryotic cells, producing the differential expression of multiple cAMP-regulated genes [68] (Table 1). The function of this enzyme leads to a disruption in the actin cytoskeleton, the inhibition of bacterial uptake by host cells, and increased endothelial permeability.

A key determinant of *P. aeruginosa*’s pathogenicity is the exotoxin (or toxin) A (ExoA or ToxA), a potent toxin secreted by Xcp-T2SS. ToxA contains four distinct structural and functional domains responsible for cell recognition, translocation of the toxin across cellular membranes, and the catalytic ADP-ribosyltransferase activity of this toxin [72]. Once internalized in host cells, ToxA catalyzes the ADP-ribosylation of the elongation factor-2 (eEF-2), resulting in the inhibition of protein synthesis and causing apoptotic cell death [73]. 

*P. aeruginosa* produces exoenzymes and exotoxins that are injected into eukaryotic cells via other secretion systems such as the type VI secretion system (T6SS). Particularly, the phospholipase D enzymes PldA and PldB, secreted by H2- and H3-T6SS, respectively, target the host PI3K (phosphoinositide 3-kinase)/Akt pathway allowing internalization of *P. aeruginosa* into epithelial cells [74,75]. This process is enhanced by the secretion of VgrG2b via H2-T6SS. VgrG2b targets microtubule components, including the γ-tubulin ring complex (γTuRC) that catalyzes microtubule nucleation, promoting a microtubule-dependent internalization of *P. aeruginosa* [76]. H2-T6SS is also used by *P. aeruginosa* to inject the Tle4 phospholipase family protein TplE into eukaryotic cells. TplE contains a eukaryotic PGAP1 (post-glycosylphosphatidylinositol attachment to proteins 1)-like domain that allows targeting the host’s endoplasmic reticulum (ER), promoting ER stress and autophagy [77]. 

Furthermore, some lipoproteins produced by *P. aeruginosa* can act as pro-inflammatory lipotoxins, causing an excessive inflammatory response in the lungs of CF patients (Table 1). This includes lipopolysaccharide transport proteins A and B (LptA and LptB, respectively) and the osmoprotective lipoprotein OsmE, which increase the production of interleukin-8 in host epithelial tissues and human macrophages [58].

### 1.4. Toxic Secondary Metabolites

*P. aeruginosa* also produces toxic secondary metabolites that induce host cell cytotoxicity (Table 1). This pathogen is notably known for the production of phenazines, which are redox-active compounds that play a major role in antibiotic resistance and virulence. Of special relevance is pyocyanin, a phenazine-derived pigment capable of inducing apoptosis in neutrophils by generating reactive oxygen species (ROS) that damage mitochondria [78]. This metabolite is crucial for *P. aeruginosa* to establish a chronic infection and to induce lung damage in mice [79]. *P. aeruginosa* also produces hydrogen cyanide, especially under low oxygen conditions and upon reaching high cell density. This toxic metabolite disrupts aerobic respiration of the host cells by binding to the Fe^3+^ of the cytochrome oxidase in the respiratory chain [80]. Additionally, its structural similarity with oxygen allows cyanide to bind to other metalloproteins, impairing a wide range of cell functions [80].

### 1.5. Iron Acquisition Systems and Factors Controlling Iron Homeostasis

Iron is an essential nutrient for all living organisms because it works as a redox cofactor of enzymes required for several vital processes. In the human body, iron is mainly found complexed with heme in hemoproteins or sequestered in the iron-storage protein ferritin inside the cell [61]. During infection, the host limits the availability of iron by inducing iron sequestration in order to combat invading microorganisms, a phenomenon known as ‘nutritional immunity’. To achieve this, the host increases the production of iron-scavenging molecules, haptoglobin, and hemopexin, which chelates free heme or hemoglobin [61]. To fulfil their iron requirements in the host, *P. aeruginosa* produces pyoverdine and pyochelin siderophores and the HasAp hemophore, which are high-affinity iron- and heme-chelating compounds, respectively [61] (Table 1). Pathogens have developed a preference for heme as iron source during the early stages of infection. Accordingly, *P. aeruginosa* possesses three different systems for heme acquisition: Phu, Has, and Hxu [81]. The TonB-dependent transporters (TBDTs) PhuR and HasR undertake transport into the periplasm of free heme and heme-HasAp complexes, respectively. The TBDT HxuA, however, plays a smaller role in heme acquisition and seems to be instead involved in the activation of virulence upon sensing heme from the host [61,81]. Pyoverdine has an extremely high affinity for iron and is even capable of stealing iron from the host’s iron-scavenging proteins, such as lactoferrin and transferrin, which confers a huge advantage to the pathogen during infection [61]. Iron competition during the host–pathogen interaction is a crucial process that can determine the success of an infection [61]. Accordingly, a pyoverdine-defective mutant *P. aeruginosa* strain shows attenuated virulence in mice and *Caenorhabditis elegans* [82,83]. Pyochelin has a lower affinity for iron than pyoverdine, but its synthesis is less energetically burdening and thus preferred when iron is moderately limited [7]. Pyochelin is known to work together with pyocyanin to induce oxidative stress and inflammation in host cells and to cause tissue damage during chronic infections by provoking a sustained inflammatory response [7].

Controlling iron homeostasis is also critical to survive within the host during infection. High amounts of intracellular iron lead to the production of reactive oxygen species (ROS) via Fenton reactions, eventually resulting in cell wall and DNA damage and lipid peroxidation [84]. In order to survive iron toxicity, bacteria have developed mechanisms to store excess of iron in heteropolymeric complexes known as bacterioferritins. In *P. aeruginosa*, bacterioferritins are composed of BfrA and BfrB subunits arranged in a spherical shell, which captures and reduces the levels of free iron in the cytoplasm [61]. *P. aeruginosa* also has a means to defend itself from oxidative damage caused by excess iron, such as superoxide dismutase SodB and catalase KatA, which target superoxide anions and hydrogen peroxide under high iron levels, respectively [61]. The response to iron toxicity is further modulated by the two small RNAs (sRNAs) PrrF1 and PrrF2, which block the production of iron storage proteins, non-essential iron-containing proteins, oxidative stress resistance, and quinolones [61,85]. These sRNAs are particularly expressed during *P. aeruginosa* infection in CF patients and were found to be required for proper infection in a murine model of lung infection [61,85].

### 1.6. Regulation of Virulence

Importantly, all these *P. aeruginosa* virulence factors are under tight regulation to prevent their production when they are not required. This is particularly important when colonizing a host to avoid alarming the immune system during the first stages of infection. Gene expression in bacteria is primarily regulated at the level of transcription initiation by modifying the affinity of RNA polymerase (RNAP) for DNA. This occurs through two main mechanisms, namely, with the substitution of the σ subunit in RNAP and with the action of transcription factors (TFs) [86]. The σ factor in RNAP is a dissociable subunit required for promoter recognition and transcription initiation. *P. aeruginosa* contains two families of σ factors, the σ^70^ and σ^54^ families, which are structurally different and use a different mechanism for transcription initiation [20]. While σ^70^ factors can perform this process on their own, σ^54^ factors require the function of ATP-dependent enhancer binding proteins (EBPs). EBPs normally consist of an N-terminal sensor domain, a conserved central AAA^+^ ATPase domain, and a C-terminal DNA binding domain [87]. Upon sensing the inducing stimulus, EBPs bind to the enhancer-binding site on target genes located upstream of the σ^54^ binding site and energize transcription initiation from σ^54^ promoters via their AAA^+^ ATPase domains. [87]. The σ^70^ family has been classified in four major phylogenetic groups based on structure and function, with Group 1 consisting of primary σ factors that triggers transcription of essential general functions and Groups 2–4 consisting of alternative σ factors (reviewed in [88,89]). The activity of RNAP loaded with primary or alternative σ factors can be modulated by TFs. TFs are DNA-binding proteins that bind to DNA regions known as operators, thereby blocking or promoting transcription initiation (reviewed in [90]). TFs control gene transcription in response to intra- or extracellular signals, and their activity is often modulated by the signal itself. Most TFs have a sensing or input domain for signal binding and an output DNA-binding domain that produces a response to the signal by modifying gene transcription. These are known as one-component systems (OCSs) and are especially abundant in bacteria that are capable of colonizing many different environments [91] like *P. aeruginosa*. TFs can be associated with sensor histidine kinases, forming the so-called two-component systems (TCSs), which are extensively present in *P. aeruginosa* [92]. Virulence in this pathogen is also controlled by quorum sensing (QS) in response to cell density, a mechanism that involves the activation of OCS proteins. This review highlights the OCSs, TCSs, and σ factors of *P. aeruginosa* that regulate the production of virulence factors during infection.

## 2. Control of *P. aeruginosa* Virulence by One-Component Systems

One-component systems (OCSs) are the simplest and most responsive transcriptional regulators of bacteria. Canonical OCSs are composed of a single cytoplasmic protein that has an input or sensor domain for signal recognition and an output DNA-binding domain that responds to the signal by modifying gene transcription (Figure 1a) [91]. One-component proteins have a wide diversity of input domains that allow the detection of multiple different signals. For the output domain, around 84% of one-component proteins have a helix-turn-helix (HTH) domain, which binds to DNA and activates or represses gene transcription [91]. However, a minority of these proteins have another type of output domain including enzymatic domains that regulate cyclic nucleotides production and protein phosphorylation. The cytosolic location of one-component proteins reduces the type of external signals that can access this bacterial compartment to be recognized by the sensor domain. For this reason, one-component systems recognize signals produced in the cytosol or small molecules able to diffuse through the membrane, like light or gases [91]. Upon signal recognition, the output domain undergoes a conformational change that induces a response, usually by exposing the HTH domain of the protein and allowing its binding to DNA [93]. The presence of a sensor and effector domain in a single protein allows for a quick response to the signal in detriment to the fine-tuned and more precise response carried out by transcriptional regulators, in which these functions reside in different proteins (e.g., two-component systems). *P. aeruginosa* harbors between 408 and 452 OCS proteins, with the type strain PAO1 encoding 423 regulators of this type [92].

### 2.1. OCSs Responding to Quorum Sensing (QS)

QS responds to cell population density, enabling bacteria to restrict the expression of specific genes to high cell densities at which the resulting phenotypes will be most beneficial. In *P. aeruginosa*, QS is performed by three autoinducers, two N-acyl homoserine lactones (AHLs), and a quinolone signal-designed PQS (from *Pseudomonas* quinolone signal) [94]. These molecules are produced by *P. aeruginosa* and diffuse freely through the membrane into the extracellular medium. When the cell density is high and the concentration of the autoinducer is above the threshold, it is recognized by a OCS protein that responds by modifying gene expression [94].

#### 2.1.1. LasR

LasR belongs to the LuxR family of regulators, which typically acts as transcriptional activators, and is the main regulator of the Las QS system in *P. aeruginosa*. LasR senses the autoinducer N-(3-oxododecanoyl)homoserine lactone (3O-C_12_-HSL), a product of the *lasI* gene, and functions as a global regulator that directly or indirectly activates the expression of over 300 genes, many of which are involved in virulence [47] (Figure 2). In addition to *lasI*, these include the elastase genes *lasA* and *lasB*, the *hcnABC* cluster for the production of hydrogen cyanide, *plcB* phospholipase, the *flp*-type IVb pili subunit, *aprA* alkaline protease, the pyoverdine regulator *pvdS*, and the *toxA* exotoxin genes [47,95] (Table 1). LasR also induces the expression of the RsaL repressor, which dampens the expression of LasR targets [96]. LasR occupies the highest position in the hierarchical QS signaling network, playing a key role activating the transcription of the Rhl and Pqs QS systems (Figure 2) [44]. Importantly, *P. aeruginosa lasR* and *lasI* mutants are considerably less virulent than the wild-type strain in a burn mouse model of infection and a rat model of acute pneumonia, and the lack of LasR produces a lower inflammatory response and impaired host colonization [44,49,97].

#### 2.1.2. RhlR

Mirroring LasR, the LuxR family regulator RhlR is the transcriptional activator of the Rhl QS system, which detects the N-butanoyl-L-homoserine lactone (C4-HSL) signal molecule generated by the product of the *rhlI* gene [98]. RhlR directly or indirectly activates the expression of over 100 genes, including several virulence genes like *rhlA* and *rhlB* for synthesis of rhamnolipids, and LasR targets *lasB*, *hcnABC*, and phenazine genes [44] (Figure 2). *P. aeruginosa rhlR* and *rhlI* mutants are considerably less virulent than the wild-type strain in murine infection models [99,100]. Despite the similarities and overlapping functions between the Las and Rhl systems, each one contributes to virulence independently, as evidenced by the synergistic effects on virulence observed when both systems are mutated [95,101]. A double *lasR rhlR* mutant resulted in lower mortality rates for the nematode *C. elegans* than single *lasR* and *rhlR* mutants [101]. Similarly, a *lasI rhlI* double mutant was more lethal than single mutants in a burn mouse model of infection [95]. Surprisingly, whilea *lasR* or *rhlR* single mutation reduces the cytotoxicity of *P. aeruginosa* towards lung epithelial cells, the cytotoxicity of the *lasR rhlR* double mutant is higher than that of the wild-type strain [101]. This suggests that some elements of the LasR and RhlR regulons may act in opposite directions [101]. 

#### 2.1.3. PqsR

The third QS regulator of *P. aeruginosa* is PqsR (also named MvfR), a transcriptional regulator of the LysR family that senses PQS, a quinole synthesized by the *pqsABCDE* and *phnAB* operons and the *pqsH* gene [34]. PqsR upregulates the expression of more than 100 genes, including the *pqsABCDE* and *phnAB* operons, and many genes of the RhlR regulon [34] (Figure 2). This overlap between these two QS systems seems to arise from PqsE thioesterase, which is not only required for PQS synthesis but also for RhlR activity [34]. Among others, PqsR activates the expression of genes encoding the virulence factors pyocyanin and hydrogen cyanide and the lectin LecA [34] (Table 1). A *pqsR* mutation reduces the mortality rate caused by *P. aeruginosa* in a burn mouse model of infection, but this effect is likely due to reduced expression of the *pqsE* gene, and therefore to the reduced RhlR activity in this mutant [34]. 

Moreover, PQS regulates the expression of many genes involved in iron acquisition such as pyoverdine and pyochelin biosynthetic genes [102]. This is likely an indirect effect due to the iron-chelating properties of PQS [61,102]. Interestingly, PqsR not only responds to PQS but also to its precursor molecule HHQ [44]. The *pqsH* gene encoding the enzyme required for the conversion of HHQ into PQS is the only gene in the biosynthetic cluster that is not regulated by PqsR but by the AraC family regulator CdpR [44,60]. According to the repressor role of this regulator, a *P. aeruginosa cdpR* mutant produced a higher mortality rate and increased lung injury and inflammation in a mouse model of acute pneumonia [60].

### 2.2. OCSs Regulating Motility, Attachment, and Biofilm Formation 

#### AmrZ

The *P. aeruginosa* AmrZ (from alginate and motility regulator) belongs to the ribbon-helix-helix family of transcriptional regulators, which mainly functions as repressors [103]. While acting as a repressor for nearly 50 genes, AmrZ can also act as an activator, controlling the expression of virulence genes such as *P. aeruginosa* alginate synthesis and transport genes encoded within the *alg* cluster [39,104] (Figure 3). Because alginate production promotes *P. aeruginosa* biofilm formation and persistence, by activating the expression of the *alg* operon, AmrZ promotes this virulence phenotype. However, in response to changes in oxygen and nutrient levels or in the presence of toxic elements such as nitric oxide, AmrZ promotes cell dispersion from bacterial biofilms by activating the expression of several hydrolase and nuclease genes such as *endA*, *pelA*, and *pslG* [105]. AmrZ also plays a major role in motility, acting as a repressor of the type IV pili precursor *pilA* and *fleQ*, a c-di-GMP effector that activates the transcription of flagellum synthesis genes [22].

### 2.3. OCSs Regulating the Production and Secretion of Extracellular Enzymes, Toxins, and Toxic Secondary Metabolites

#### 2.3.1. ExsA

ExsA belongs to the AraC family of regulators that mostly act as transcriptional activators upon signal sensing [106]. This regulator induces the expression of *P. aeruginosa* T3SS in response to low calcium levels, recruiting the RNAP complex to the promoter of the T3SS cluster [55]. This results in transcription from the 11 promoters controlling T3SS expression and secretion of the ExoT, ExoS, ExoU, and ExoY effectors (Table 1). ExsA was recently found to also activate the transcription of the *impA* gene that specifies a metalloprotease secreted by T2SS, which confers resistance to phagocytosis from macrophages [107]. In contrast to most AraC regulators that are activated by a small-molecule ligand, ExsA is part of the ExsACDE signaling cascade [55]. In non-induced conditions, ExsA is sequestered by ExsD, and the chaperone ExsC forms a complex with ExsE. Contact with host cells causes T3SS to secrete ExsE, switching ExsC to pair with ExsD instead. ExsA, now free from ExsD repression, activates gene expression [55]. Cytotoxicity assays showed that the cytotoxicity of a *P. aeruginosa exsA* mutant toward different cell lines was severely impaired [108,109]. Moreover, an *exsA* mutant is more susceptible to phagocytosis than the wild-type strain, likely due to the ExsA-dependent transcription of *impA* [107,110].

#### 2.3.2. Sfa2

The Sfa2 protein is an EBP that activates the σ^54^-dependent transcription of *P. aeruginosa* H2-T6SS. Sfa2 binds to the enhancer-binding site on H2-T6SS genes, promoting the transcription of ~15 genes in this gene cluster, which includes the *sfa2* gene itself [111]. A paralogue EBP regulator known as Sfa3 is encoded within the *P. aeruginosa* H3-T6SS gene cluster, suggesting a similar control of H3-T6SS expression by σ^54^ and this EBP [112].

#### 2.3.3. GbdR

The AraC regulator GbdR activates the expression of hemolytic phospholipase C PlcH upon sensing glycine betaine and dimethylglycine [45]. PlcH produces phosphorylcholine, which is then dephosphorylated by PchP to release choline. Choline metabolism leads to the production of glycine betaine, which can either accumulate as an osmoprotector and a bacterial surfactant or be used as a nutrient source. GbdR acts as a switch for the transition between both fates upon accumulation of glycine betaine by promoting the expression of genes involved in choline conversion into serine and pyruvate [45]. Expression of the *gbdR* gene is activated by σ^RpoN^ and the EBPs CbrB and NtrC in the absence of carbon/nitrogen sources, and it is repressed by the *betBA* cluster repressor BetI in the absence of choline [113]. Although the role of GbdR in virulence has not been assayed to date, choline metabolism was shown to be important for the proper colonization and survival of *P. aeruginosa* in a mouse model of lung infection [19].

#### 2.3.4. PvrA

The *Pseudomonas* virulence regulator PvrA belongs to the TetR family of transcriptional regulators that normally act as repressors of gene expression. Like GbdR, PvrA promotes the expression of the *plcH* phospholipase gene, although in response to fatty acyl-CoAs such as palmitoyl-CoA [52,114]. PvrA serves as a link between virulence and metabolism of phosphatidylcholine and long-chain fatty acids, one of the main carbon sources for *P. aeruginosa* during lung infection and a major component of the lung surfactant. In addition to directly controlling the expression of several metabolism genes and *aprA* alkaline protease, PvrA activates the production of pyocyanin and rhamnolipids by promoting transcription of the PQS biosynthesis operons of *pqsABCDE* and *phnAB* [52,114]. Moreover, it represses the expression of *phaG* encoding a protein involved in the conversion of 3-hydroxyacyl-acyl carrier protein (ACP), the substrate required for rhamnolipids production, into polyhydroxyalkanoate energy storage compounds [52]. In this way, PvrA acts as a switch diverting 3-hydroxyacyl-ACP use toward rhamnolipids production instead of energy storage. Accordingly, a mutation of *pvrA* was shown to significantly reduce the colonization and survival of *P. aeruginosa* in the lungs of mice [52,114].

#### 2.3.5. SphR

SphR is an AraC family regulator that responds to sphingosine and activates the expression of genes that *P. aeruginosa* uses to feed on sphingolipids from the host cell membranes and the surfactant of mammalian lungs [54,115]. SphR activates the expression of the sphingolipid metabolism genes *sphA* and *sphBCD* and the ceramidase gene *cerN* [54,115]. In addition to enhancing the activity of PlcH phospholipase (Table 1), CerN is involved in the conversion of ceramide into sphingosine and fatty acids. *P. aeruginosa* uses fatty acids as a nutrient source, while sphingosine is used to attack host cells by disrupting the skin barrier [54]. SphR was found to be important for *P. aeruginosa* survival inside the lungs of mice, especially to overcome the antimicrobial effects of sphingosine [54,115].

#### 2.3.6. PtxR

PtxR belongs to the LysR family of regulators, generally regarded as transcriptional activators of their target genes but with repressor activity on their own genes [116]. PtxR activates the transcription of the exotoxin A gene *toxA* and the *lasB* elastase and represses the transcription of *rhlA* rhamnolipids and *phzA1* phenazine genes [43]. The activity of PtxR is modulated by the 2-ketogluconate sensor PtxS, which always acts as a repressor of gene expression regardless of PtxR’s role [117]. When PtxR acts as a repressor, PtxS binds to both DNA and PtxR, creating a DNA loop that promotes repressor activity. Repression is relieved by the presence of 2-ketogluconate, which releases both proteins from DNA. Conversely, when PtxR acts as an activator, PtxS binds to PtxR–DNA complexes, preventing gene expression. Upon detection of 2-ketogluconate, PtxS dissociates from PtxR, which activates gene expression [56,117].

#### 2.3.7. SoxR

SoxR belongs to the MerR family of transcriptional regulators, which typically act as activators of gene expression [118]. SoxR regulators normally respond to oxidative stress by sensing superoxide; however, in *P. aeruginosa*, SoxR is activated by pyocyanin in a superoxide-independent manner and controls the release of phenazines and other chorismate-derived compounds [119,120]. To date, only two gene clusters have been identified as being part of the SoxR regulon: monooxygenase PA2274 and the multidrug efflux pump operon *mexGHI-opmD* [119,121]. The efflux pump MexGHI-OpmD was shown to increase the resistance to antibiotics such as norfloxacin and tetracycline, and it is involved in the export of a phenazine precursor required for pyocyanin production [122,123]. This efflux pump is also required for the production of virulence factors such as LasB elastase, rhamnolipids, pyocyanine, and pyoverdine, probably caused by the genetic response triggered by phenazines as signal molecules [122]. Accordingly, a *P. aeruginosa soxR* mutant was found to be attenuated for virulence in pulmonary and burn mouse models of infection [121,124].

### 2.4. OCSs Regulating the Acquisition and Homeostasis of Iron

#### 2.4.1. Fur

The ferric uptake regulator Fur is the main regulator of iron homeostasis in *P. aeruginosa*. This regulator facilitates host infection by maintaining sufficient intracellular iron levels while avoiding iron toxicity. Upon binding iron, Fur directly represses the expression of iron acquisition genes and indirectly activates the expression of iron storage and detoxification genes, like the bacterioferritin gene *bfrB*, superoxide dismutase *sodB*, and catalase *katA* [61,62] (Table 1). Fur expands its regulation of iron homeostasis by repressing the expression of two sRNAs, i.e., PrrF1 and PrrF2, which block the production of over 50 proteins by binding their mRNA targets [61,85]. Fur represses the expression of pyoverdine and pyochelin biosynthesis and transport genes, which are thus only produced under iron-limited conditions [61]. Expression of the hemophore HasAp and the Phu, Has, and Hxu heme transport systems is also repressed by Fur under iron sufficient conditions. Moreover, Fur also represses the expression of other virulence factors including exotoxin A, the PIV/PrpL and IcmP proteases, extracellular LipA lipase, and hemolytic PlcH phospholipase C [50]. Fur is a global regulator that controls the expression of hundreds of genes involved not only in iron acquisition and virulence but also in some other aspects of *P. aeruginosa* biology such as respiration, metabolism, and stress responses [50].This wide spectrum of Fur targets explains why this protein is essential in *P. aeruginosa* for growth on solid media, especially considering the central role it plays in iron homeostasis [61,62].

#### 2.4.2. PchR

PchR belongs to the AraC family of transcriptional regulators and activates the transcription of gene clusters *pchDCBA* and *pchEF* for pyochelin synthesis and *fptABCX* for pyochelin uptake [61]. The iron-loaded form of pyochelin acts as an effector molecule, activating the gene transcription of PchR targets, while PchR represses its own expression when siderophore is absent [61]. Pyochelin was shown to enhance growth and lethality of *P. aeruginosa* during infections, and a mutant defective in pyochelin and pyoverdine production was shown to be less virulent than a single-pyoverdine mutant [7]. In addition to pyochelin production, PchR controls the expression of the *pqsABCDE* gene cluster for PQS production and a number of virulence factors related to QS, such as the elastase *lasB* gene, the lectin *lecA* and *lecB* genes, and the gene clusters *phzA1-G1* and *hcnABC* for phenazine and hydrogen cyanide production, respectively [28]. PchR is also involved in regulating the expression of other virulence-related genes, such as the *algD* gene for alginate production, the *fimV* gene for the synthesis of type IVa pili, and the bacterioferritin *brfB* gene [28].

#### 2.4.3. AmpR

The β-lactam resistance regulator AmpR belongs to the LysR family of transcriptional regulators, and it is an important regulator of iron acquisition and the oxidative stress response [59]. AmpR acts as a global regulator, controlling the expression of more than 500 genes [59]. AmpR promotes iron acquisition by activating the transcription of regulatory anti-sense sRNA asPrrF1 (antagonizing the sRNA PrrF1), as well as the pyoverdine and pyochelin biosynthesis genes and the hemophore *hasAp* gene [59]. The oxidative stress response is mediated via AmpR by the activation of the *katA* catalase gene and the hydrogen peroxide resistance *rgRgsA* sRNA [59]. Consequently, an *ampR* mutation results in impaired growth under iron-limiting conditions and under H_2_O_2_-induced oxidative stress [59]. Moreover, AmpR is critical for *P. aeruginosa* to survive treatment with β-lactam antibiotics, and it also contributes to increasing *P. aeruginosa*’s “evolvability”, raising the chances of strengthening or developing new antibiotic resistances [125]. Under physiological conditions, AmpR is repressed by the cell wall precursor UDP-MurNAc pentapeptide. When β-lactam antibiotics cause cell wall damage, the muropeptides resulting from its restoration are internalized and bind AmpR to activate the transcription of the *ampC* β-lactamase gene [126]. Upon activation, AmpR promotes the expression of other key virulent determinants such as the HHQ transporter *mexEF-oprN* genes, the *phz* and *hcn* gene clusters for the production of pyocyanin and hydrogen cyanide, respectively, and several genes from T3SS and T6SS [59,127]. Accordingly, a *P. aeruginosa ampR* mutant was found to reduce mortality in the *C. elegans* infection model [127].

## 3. Control of *P. aeruginosa* Virulence by Two-Component Systems

Two-component systems (TCSs) are one of the main pathways for sensing and responding to environmental cues in prokaryotes, which are also present in eukaryotes [27]. Contrary to OCSs, the sensor and effector domains of TCSs are part of two different proteins. In a canonical TCS, the sensor protein is a membrane-embedded histidine kinase (HK) with the sensor domain facing outside the cytosol, which, upon signal recognition, phosphorylates a response regulator (RR) that mediates the response (Figure 1b) [128]. Next to the sensor and the transmembrane domains, HKs have a highly conserved cytosolic core kinase domain that, upon signal sensing, binds ATP and autophosphorylates in a conserved His residue [128]. Next, a phosphate transference reaction from the HK to its cognate RR occurs. Most RRs possess two main domains: a conserved N-terminal regulatory domain that receives the phosphate group from the His of the HK in an Asp residue and a variable C-terminal effector domain that generally binds to DNA [128]. The RR can act as a transcription activator or repressor, and it is common that the RR induces the transcription of the genes encoding a TCS [129]. The half-life of RR-P can range from seconds to hours, and its activity is turned off by phosphatases that hydrolyze phosphate [128]. Many RRs have autophosphatase activity that decreases the lifetime of the phosphorylation, and several HKs also possess phosphatase activity and are able to dephosphorylate their cognate RRs [128]. Important variations from this archetypal model of TCSs have evolved over time. The most outstanding modification is the existence of kinases with numerous phosphodonor sites [128]. The succession of several phosphorelay steps facilitates an accurate control of the pathway activation and reduces the occurrence of non-specific cross-talk because of the presence of several regulatory checkpoints [128]. The genome of *P. aeruginosa* PAO1 encodes 41 HKs and 68 RRs, having an additional 18 hybrid proteins exhibiting domains from both types of proteins [92]. Although TCSs in this pathogen have been studied for a long time, the specific signals activating these pathways have been rarely identified, and several remain unknown.

### 3.1. TCSs Regulating Motility, Attachment, and Biofilm Formation

#### 3.1.1. PilS-PilR

The TCS PilSR is the main regulator of the *pilABCD* cluster controlling type IVa pili motility [25]. The HK PilS interacts with the pilin subunit PilA, detecting changes in its abundance and controlling its production. When PilA is not detected, this results in a higher activity of the RR PilR in order to replenish the PilA pool in the cell. RNAseq analyses using a *pilA* mutant, in which PilR was constitutively active and a *pilR* mutant, allowed the identification of 10 genes that were inversely regulated in both strains, including the *hcpA* and *hcpB* effectors for T6SS [21]. In addition to controlling type IVa pili motility, the PilSR system plays a role in swimming motility by regulating the expression of the flagellum TCS FleSR [21].

#### 3.1.2. FleS-FleR

FleSR activates the transcription of the basal body, hook, and filament proteins of flagella, and it is thereby important for swimming motility [48]. Additionally, this TCS enhances *P. aeruginosa* cytotoxicity because it promotes the expression of *toxA* exotoxin, *lasB* elastase, and T3SS, likely via the ExsA regulator [48]. However, it is not clear how FleSR controls AmrZ, as no direct binding of the RR to the promoter has been observed [130]. Expression of the *fleSR* gene cluster depends on the diguanylate effector FleQ to control its transcription in a c-di-GMP dependent manner as well as on the TCS PilSR [21,131].

#### 3.1.3. RocS-RocR-RocA1

The HK RocS controls the expression of the *cupB* and *cupC* systems (Table 1) by interacting with two different RRs: the activator RocA1 and the repressor RocR [29,132]. RocA1 activation follows the classical TCS phosphorylation-dependent mechanism, with phosphorylated RocA1 acting as a transcriptional activator of the *cupB* and *cupC* gene clusters [133]. RocR repression of the *cupB* and *cupC* systems depends on c-di-GMP levels, and this regulator contains a C-terminal EAL domain to sense and catalyze this signal molecule [133]. It has been suggested that the activation of c-di-GMP phosphoesterase activity of RocR takes place upon phosphorylation by RocS and that gene repression may happen by competition with RocA1 for the HK [133].

#### 3.1.4. RcsC-RcsB and PvrS-PvrR

The TCSs RcsCB and PvrSR control the expression of the *P. aeruginosa* PA14 strain-specific *cupD* gene cluster, with their RRs RcsB and PvrR acting as an activator and a repressor, respectively, of the transcription of this system [30]. In the most recent model, the HK RcsC transfers a phosphate group from the HK PvrS to the RR RcsB [30]. Phosphorylated RcsB activates the transcription of *cupD*, as well as other target genes such as the DNA-binding PA3714 and PA0034 proteins [30]. RcsC does not only mediate in the phosphotransference between PvrS and RcsB, but it also has an inhibitory role as a phosphatase to inactivate RcsB [30]. The remaining RR, PvrR, is a c-di-GMP-dependent repressor of the *cupD* cluster, with a C-terminal EAL domain similar to the RocR repressor, suggesting a similar mechanism for PvrR repression. [30]. All four proteins of this intertwined signaling network are required for *P. aeruginosa* virulence because single mutations in any of these genes were found to considerably reduce mortality in a burn mouse model of infection [134].

#### 3.1.5. PprA-PprB

The *cupE* cluster is regulated by the *Pseudomonas* permeability regulator PprAB [11]. The RR PprB induces the expression of the *cupE* operon and several genes from the type IVb pili apparatus, such as the *tad* locus, the *flp* pilin, and the adhesin *bapABCD* gene cluster [12]. PprAB also indirectly affects the expression of the PQS system, thus having a broader impact on virulence [12]. Because of this, overexpression of the PprB protein was found to cause a hyper-biofilm phenotype, higher susceptibility to antibiotics such as tobramycin due to a higher membrane permeability, and attenuated virulence in a fruit fly infection model [12].

#### 3.1.6. FimS-AlgR

FimS-AlgR controls alginate production via the activation of the *algC* gene and *algD* operon for alginate biosynthesis and transport [135] (Figure 3). This TCS also induces expression of the *fimU* operon involved in type IVa pili motility and the QS-related genes *rhlI*, *rhlA*, and *hcnA* [135]. The HK FimS interacts with the surface contact sensor PilJ, which senses tension changes in the pili via the PilA protein-activating FimS, which in turn phosphorylates the RR AlgR [136]. AlgR can function as a repressor or an activator of gene transcription depending on its phosphorylation state, which seems to depend on the stage of infection. In early stages of biofilm formation, AlgR is phosphorylated and functions as an activator favoring motility and cell attachment. Unphosphorylated AlgR acts as a repressor and is more important in late-stage biofilms. AlgR also promotes *mucR* expression, a diguanylate cyclase that alters c-di-GMP levels, thus promoting biofilm formation and the establishment of chronic infections [135]. Mutations in either *fimS* or *algR* result in a nearly complete loss of twitching motility [135]. The *fimS* or *algR* mutants were found to display similar killing rates as the wild-type, but a phosphomymetic mutant of AlgR showed delayed killing in a *C. elegans* model [137].

#### 3.1.7. KinB-AlgB

The TCS KinB-AlgB also controls alginate production by activating the expression of the alginate biosynthetic cluster (Figure 3) [40]. The unphosphorylated form of the RR AlgB binds to three sites on the *algD* promoter, activating its transcription. Mutation of the HK *kinB* gene results in alginate overproduction due to the constitutive activation of AlgB that, in this mutant, cannot be inactivated by phosphorylation [40]. This mutation also causes defects in the acute phase of infection, producing fewer QS-regulated virulence factors, such as pyocyanin and elastase, and showing impaired swimming motility and biofilm formation [138]. KinB has both kinase and phosphatase activity, playing a dual role in the phosphorylation of AlgB. In the phosphorylated form, AlgB represses acute virulence and alginate production. Dephosphorylation of this protein by KinB promotes these phenotypes during the acute infection stage [139]. While a *kinB* mutation was found to decrease *P. aeruginosa* virulence in zebrafish embryo and murine models of infection, *algB* single- or *kinB*/*algB* double-mutants produce similar mortality rate compared to the wild-type strain [138,139,140].

#### 3.1.8. GacS-GacA

The GacSA TCS governs switching from acute to chronic *P. aeruginosa* infection by activating the transcription of the *rsmY* and *rsmZ* genes. These two sRNA genes are the only targets identified for this TCS, and they modulate the activity of the global regulator RsmA (Figure 4) [24]. RsmA is a RNA-binding protein that blocks the translation of genes for biofilm formation such as the *pel* and *psl* EPS genes, while indirectly activating transcription of motility and virulence factors such as the type IV pili and T3SS genes [24]. RsmA is thus one of the main proteins controlling the transition from a planktonic to a sessile lifestyle, which correlate with the acute and chronic stages of infection, respectively. The sRNAs *rsmY* and *rsmZ* bind to RsmA, preventing the binding to its targets, and thus promoting biofilm formation and the development of a chronic infection [24].

The activity of the HK GacS is modulated by two additional HKs, RetS and LadS (Figure 4) [141,142]. RetS is a dimeric HK that binds to a GacS homodimer to form a heterocomplex that represses GacS activity, thereby promoting motility and virulence. The HK PA1611 seems to intercede in the formation of this heterocomplex by forming a heterodimer with RetS (Figure 4). LadS is a HK that can transfer a phosphate to GacS, activating its signaling cascade and promoting biofilm formation while attenuating virulence [142,143]. Whereas no signal has been yet identified for GacS, calcium seems to be the inducer of the LadS-dependent activation of the GacSA cascade, and mucins and associated glycans (normally covering epithelial cells) are the signals sensed by RetS to inhibit GacS [142,144].

The *P. aeruginosa* PA14 strain harbors a truncation in the LadS HK, which seems to partially account for the increased cytotoxicity of this strain compared to other strains such as PAO1 [141]. Virulence assays using the *Galleria mellonella* larvae model of infection showed that a *gacS* deletion did not have a significant effect on virulence. However, a deletion of the HAMP domain, transducing the signal from the periplasmic sensor to the cytoplasmic kinase domain, resulted in an important drop in mortality [145]. Likewise, deleting the HK gene *retS* increased larvae survival, implying that either the HAMP domain of GacS is important for RetS inhibitory activity or a mutant in the HAMP domain results in hyperactive GacS [145].

#### 3.1.9. BfiS-BfiR, BfmS-BfmR, and MifS-MifR

The TCSs BfiSR, BfmSR, and MifSR act sequentially in the irreversible attachment and maturation stages 1 and 2 of the biofilm formation process [27]. BfiSR regulates irreversible attachment by activating the transcription of the RNAse G *cafA* gene, which reduces *rsmZ* levels, thereby enhancing RsmA activity (Figure 4) [27]. The BfiS HK also interacts with the hybrid HK SagS, which seems to act by modulating the phosphorylation state of BifS [146]. A *bfiS* mutant was found to reduce mortality in a plant infection model, while mutation of its regulatory kinase SagS produced a hypervirulent phenotype [146]. Furthermore, BfmSR regulates biofilm maturation by activating the expression of the prevent-host-death gene *phdA*, thus suppressing the premature release of eDNA [14]. The BfmR RR controls the expression of other transcriptional regulators. BfmR represses the expression of the QS *rhlR* gene and the Zn^2+^ sensing *czcRS* TCS, controlling phenazine production, and induces the expression of the T3SS regulator *exsA* [147,148]. The virulence of a *bfmS* mutant was found to be severely impaired, while the virulence of a double-*bfmSR* mutant was similar to the wild-type strain, probably by relieving BfmR repression on virulence regulators [147]. Lastly, the MifSR TCS affects microcolony and cluster formation during the last stages of the biofilm maturation process. Upon sensing α-ketoglutarate, the HK MifS phosphorylates MifR, which acts as an EBP, assisting the σ^54^ factor RpoN to bind to the promoter region of α-ketoglutarate transport genes [27,149]. α-ketoglutarate is a preferred carbon source for *P. aeruginosa*, and it is required for virulence in pathogenic bacteria [149]. MifR has also been shown to alter the production of at least 18 proteins when mutated or overexpressed, many of which are involved in growth under anaerobic conditions, suggesting a link between biofilm formation and oxygen restriction [42]. In a pneumonia mouse model, a *mifR* mutant showed significant defects in virulence and mortality, with fewer lung injuries, a lower inflammatory response, and reduced cell death [150].

#### 3.1.10. GtrS-GltR

The glucose uptake and metabolism TCS GtrS-GltR controls *rsmY* and *rsmZ* levels and, thereby, has an effect on the RmsA-mediated phase switch and virulence [26]. By inducing the expression of the *oprB* gene for glucose uptake, pyruvate metabolism is activated, which leads to a repression of *rsmY* and *rsmZ* expression [26]. In a mouse model of acute pneumonia and a fruit fly infection model, a *P. aeruginosa gtrS* mutant was impaired for colonization and dissemination and had reduced mortality, mirroring the virulence traits of a *rsmA* mutant [26,151]. Activation of the HK GtrS depends on the binding to 2-ketogluconate or phosphogluconate, although it seems to also respond to glucose via the periplasmic protein GltB [57,151]. GltR represses the expression of several genes that are essential for metabolism, such as the *glk* and *edd*/*gap-1* genes, and for virulence, such as the exotoxin *toxA* gene [57,151].

### 3.2. TCSs Regulating the Production and Secretion of Extracellular Enzymes, Toxins, and Toxic Secondary Metabolites

#### PhoR-PhoB

Under phosphate limitation, the TCS PhoRB activates the expression of genes involved in phosphate assimilation and metabolism, including the *hxc*-T2SS operon and the *pstSCAB-phoU* transport system [152,153]. Hxc-T2SS acts as an alternative system to scavenge inorganic phosphate under phosphate-limiting conditions by secreting alkaline phosphatases to the extracellular space [154]. Phosphate depletion has shown to dramatically increase virulence in *P. aeruginosa* both in surgical injuries and animal infection models and, conversely, hosts treated with phosphate supplements have shown to be less susceptible to *P. aeruginosa* infections [155,156]. When phosphate conditions are favorable, the PhoU subunit of the Pst transport complex interacts with the HK PhoR to prevent its autophosphorylation [152,153]. In response to low phosphate, PhoR phosphorylates the PhoB RR, which, in addition to genes involved in phosphate transport and metabolism, activates the transcription of H2- and H3-T6SS and the three QS systems [157]. Consequently, PhoB promotes pyocyanin production and internalization by epithelial cells in low-phosphate conditions [157]. It was found that clinical isolates from *P. aeruginosa* with a PhoB-dependent formation of PstS rich structures resulted in a hypervirulent phenotype, causing the death of 60% of surgically injured mice. Phosphate depletion during *P. aeruginosa* infection of *C. elegans* results in the so-called “red death”, a lethal phenotype resulting from the combined activation of PhoB, PqsR, and the pyoverdin acquisition systems [156].

### 3.3. TCSs Regulating the Acquisition and Homeostasis of Iron

#### BqsS-BqsR

The TCS BqsSR responds to extracellular Fe^2+^ and activates its own expression by triggering the expression of the *bqsPQRST* cluster and that of spermidine, ferric reductases, carbonic anhydrases, and sulphate transporter genes [158,159]. This TCS promotes the synthesis of rhamnolipids and pyocyanin by increasing the production of C4HSL and PQS signaling molecules [158,159]. The reduced Fe2+ form activating BqsSR is scarce in the oxygenic environments that conform the usual habitats of P. aeruginosa, where iron is usually found as Fe3+. However, it was observed that Fe^2+^ locally accumulates in microenvironments in the lungs of CF patients and that it further accumulates by the increase in cell density and phenazine production, providing a link between QS, iron acquisition, and virulence [158,160].

## 4. Control of *P. aeruginosa* Virulence by Sigma (σ) Factors

Contrary to OCSs and TCSs, which modulate transcription by binding to operator sequences, σ factors affect gene expression by directly recruiting the RNAP core enzyme to their promoter regions and then starting transcription. As is customary for most bacterial species, *P. aeruginosa* produces a single σ^54^ factor known as σ^RpoN^. This σ^54^ factor recognizes the promoter region of hundreds of genes with the assistance of up to 22 different EBPs (depending on the strain) (e.g., the OCS Sfa2 or the RRs AlgB, PilR, or FleR) [20,111,140]. Conversely, bacteria typically produce several σ factors of the σ^70^ family. Alternative σ factors promote the transcription of functions required only under specific conditions, and their activity is controlled by specific stimuli. This control can be exerted at different levels (i.e., by modulating expression, covalent modifications, proteolysis), although it is particularly common the post-translational control of alternative σ factors by an anti-σ factor that prevents the interaction of the σ factor with the DNA and RNAP core enzyme in absence of the stimulus (Figure 1c). *P. aeruginosa* produces one primary σ factor encoded by the *rpoD* gene and between 22 and25 alternative σ factors, depending on the strain. Of those, Group 4, containing the so-called extracytoplasmic function σ (σ^ECF^) factors, is the most numerous and diverse group, with *P. aeruginosa* encoding between 19 and 21 σ factors in this group [161].

### 4.1. σ Factors Regulating Motility, Attachment, and Biofilm Formation

#### 4.1.1. σ^FliA^ (σ^28^)

The σ^FliA^ factor controls the expression of the flagella biosynthesis genes of *P. aeruginosa* and is thus essential for bacteria motility. Its activity is modulated by its cognate anti-σ factor FlgM via partner switching with the protein HsbA, which acts as an anti-σ factor antagonist when unphosphorylated [162]. The phosphorelay system leading to HsbA phosphorylation is intertwined with the RsmA signaling network, acting as a switch between a motile planktonic lifestyle when unphosphorylated and biofilm formation when phosphorylated [20,162]. In addition to being essential for flagella-driven swimming motility, σ^FliA^ is important for swarming motility [163]. A transcriptomic study on a *P. aeruginosa fliA* mutant of the PAO1 strain showed that several effector genes of Xcp-T2SS (i.e., *lipC* lipase and *phlC* phospholipase C), H2-T6SS (i.e., *vgrG*, *hcp*, *clpV2*, and *lip2*), and T3SS (i.e., *exsC*, *exsE*, and *pscBCD*) were strongly downregulated, showing that this σ factor is required for the expression of these virulence factors [53]. However, σ^FliA^ represses the expression of other virulence determinants such as pyocyanin [53]. Interestingly, a *fliA* mutant of the hypervirulent *P. aeruginosa* B136–33 strain showed less capacity for adhesion to Caco-2 monolayer cells and a lower invasion activity in MDCK cells, and its capacity to colonize mice was considerably reduced [53]. σ^FliA^ is essential for successful adhesion and infection of H9C2 cardiomyocyte cells of the PAO1 strain, which further supports the crucial role of flagella in pathogenesis through motility [164].

#### 4.1.2. σ^SigX^

The σ^SigX^ factor regulates the attachment and adherence of *P. aeruginosa* to host tissues by controlling the swarming and twitching motility of this bacterium [23,165]. It was proposed that σ^SigX^ activity is controlled by a mechanosensing complex involving the anti-σ factor CfrX, the outer membrane porin OprF, and the ion channel CmpX [166]. The lack of this σ factor decreases the expression of *pilA* and *flp* genes, encoding the pilin subunits of the type IVa and IVb pili, respectively, and the *tad* operon, encoding the machinery that assembles type IVb pili [23]. Furthermore, the expression of the RR gene *pprB*, which promotes the transcription of the *flp* type IVb pilin, the *bap* adhesion system genes, and the *cupE* CUP pili cluster, was found to be downregulated in a *sigX* mutant [23]. Consequently, the lack of σ^SigX^ reduces the biofilm formation capacity of *P. aeruginosa* [23]. A *sigX* mutation impairs the production of other virulence factors, like exotoxin A, pyocyanin, the T1SS-related genes *aprA*, *aprD*, and *aprX*, Xcp-T2SS, and H3-T6SS [23]. Accordingly, a *P. aeruginosa sigX* mutant was found to be less cytotoxic towards A549 epithelial pulmonary cells and Caco2/TC7 intestine cells than the wild-type strain [23,165].

Furthermore, σ^SigX^ regulates the expression of several fatty acid biosynthesis genes, thus playing a crucial role in membrane fluidity and homeostasis [166]. Importantly, many genes of the σ^SigX^ regulon are associated with cell membrane integrity, thus suggesting that σ^SigX^ could regulate the response of *P. aeruginosa* to membrane-disrupting antimicrobials. Interestingly, a *sigX* mutant strain showed considerably higher resistance to the antimicrobials tetracycline, nalidixic acid, and erythromycin [23].

Finally, σ^SigX^ also plays a key role in regulating iron uptake by promoting the expression of the *prrF1* sRNA gene, which represses iron sequestration under iron-starvation conditions [51,167]. Furthermore, the absence of σ^SigX^ was found to cause a reduced expression of the pyochelin biosynthesis genes and of several TBDTs involved in iron (ChtA, FpvA and FpvB) and heme uptake (PhuR) [23,168].

#### 4.1.3. σ^AlgT^

The σ^AlgT^ factor is the main regulator of the *alg* operon encoding the biosynthetic pathway of the EPS alginate, a main component of the extracellular matrix of *P. aeruginosa* biofilms [169] (Figure 3). This σ factor, which is a homologue to the stress-responsive σ^E^ factor of *E. coli*, is kept inactive under non-stress conditions by the anti-σ factor MucA encoded within the *algT-mucABCD* operon [161]. The presence of unfolded proteins leads to the activation of σ^AlgT^ via regulated intramembrane proteolysis (RIP) by the DegG and RseP proteases (Figure 3) [161]. The main cause of the mucoid phenotype of *P. aeruginosa* is a mutation of the *mucA* gene, which results in the constitutive activation of σ^AlgT^ and continuous production of alginate [170]. In addition to activating alginate production, σ^AlgT^ controls biofilm formation by modulating the expression of the of the *psl* EPS gene. Basal σ^AlgT^ concentrations activate the expression of *psl*, while high σ^AlgT^ concentrations induce the transcription of the biofilm inhibitor *rsmA* gene, which prevents the production of the Psl EPS [41,169]. σ^AlgT^ also activates the expression of *lecB*, which codifies for a lectin involved in biofilm formation and post-transcriptional type IVa pili biogenesis [33]. Furthermore, σ^AlgT^ affects the attachment to solid surfaces via induction of the paerucumarin *pvcA* gene, which enhances expression of the *cup* adhesion genes [171].

σ^AlgT^ also regulates the expression of several other virulence determinants such as the LasB and the AprA proteases and the *hcn* hydrogen cyanide biosynthesis genes [46,172]. Several genes encoding pro-inflammatory lipoproteins (i.e., *lptA*, *lptE*, and *osmE*) are induced in mucoid *P. aeruginosa* strains because their expression is promoted by σ^AlgT^, which causes excessive inflammation in CF patients [58,172]. Moreover, the genes *slyB*, encoding a putative porin involved in membrane integrity, *osmC*, encoding a peroxiredoxin inducible by osmotic stress, and *pfpI*, encoding a protease that protects against oxidative stress, are differentially expressed in mucoid *P. aeruginosa* strains in a σ^AlgT^-dependent manner [46,166,172]. Furthermore, σ^AlgT^ regulates the expression of the heat-shock response σ^RpoH^ factor. The regulation of all these genes related to adaptation against several stresses by σ^AlgT^ provides an advantage for this pathogen that can overcome the host immune response during infection.

The σ^AlgT^ regulon includes a wide assortment of regulatory genes, reflecting the complex σ^AlgT^ signaling network. This network includes several OCSs and TCSs, such as the alginate regulators AmrZ, KinB/AlgB, and FimS/AlgR as well as σ^ECF^ factors such as σ^SbrI^ involved in motility and biofilm formation [51]. σ^AlgT^ indirectly controls motility by inducing *amrZ* expression, which, subsequently, represses *fleQ*, encoding for the major regulator of flagella biosynthesis [22]. In a *mucA* mutant strain, σ^AlgT^ activation was found to cause downregulation of the virulence factor regulator gene *vfr* via the RR AlgR [173].

#### 4.1.4. σ^RpoH^ (σ^32^)

The σ^RpoH^ factor is responsible of the *P. aeruginosa* response to heat-shock and accumulation of misfolded proteins [20]. σ^RpoH^ is essential for viability in *P. aeruginosa* and induces the expression of several virulence factors including the *mucA* and *mucB* genes, which modulate the activity of the σ^AlgT^ factor, and the global regulator RsmA controlling biofilm formation [38]. Furthermore, this σ factor positively regulates the expression of several chaperones, proteases, and accessory proteins, such as ClpX, ClpB, FtsH, and GrpE, that are required for refolding or degradation of non-functional proteins [20,38]. Overproduction of σ^RpoH^ significantly increases the sensitivity of *P. aeruginosa* to gentamicin and other aminoglycoside antibiotics such as tobramycin in biofilms [38]. Interestingly, a transcriptome study on *P. aeruginosa* biofilms in a mouse wound model showed that much of the heat-shock response genes regulated by σ^RpoH^ are induced in vivo during infection [174].

#### 4.1.5. σ^RpoS^ (σ^38^)

The σ^RpoS^ factor is responsible for regulating the global stress response of *P. aeruginosa*. The activity of this alternative σ factor is induced during the stationary phase and modulates the expression of more than 700 genes of many functional categories [175]. σ^RpoS^ directly regulates the expression of the RR PprB which, in turn, promotes the expression of several cell adhesion and motility-related genes, as described before [31]. Moreover, σ^RpoS^ positively regulates biofilm formation by promoting the expression and production of the EPS Psl and alginate [36,37]. Surprisingly, the *rpoS* mutation was found to cause a hyperbiofilm phenotype and heightened virulence of *P. aeruginosa* in a macrophage cytotoxicity model [176]. This hyperbiofilm phenotype could be caused by the increase in c-di-GMP observed in *P. aeruginosa rpoS* mutants [31]. The expression of *rpoS* is highly induced in *P. aeruginosa* isolates from CF patients [177]. In accordance, a *P. aeruginosa rpoS* mutant was found to increase virulence and cause more lung damage than the wild-type strain in a rat chronic-infection model [36].

#### 4.1.6. σ^SbrI^

The σ^ECF^ factor SbrI is under the control of the anti-σ factor SbrR and controls swarming motility and biofilm formation in *P. aeruginosa* [35]. σ^SbrI^ promotes the transcription of the *muiA* (from mucoidy inhibitor A) gene, which encodes the MuiA protein that inhibits alginate production by interfering with the P_algD_ promoter activity [35]. However, the phenotypic characterization of a *sbrR* mutant, which leads to constitutive activation of σ^SbrI^, showed that this σ factor inhibits swarming motility and enhances biofilm formation, suggesting that the role of MuiA as an inhibitor of alginate production and biofilm formation should be re-evaluated [35]. σ^SbrI^ and MuiA are overexpressed under osmotic shock conditions, after treatment with the cell-wall inhibitory antibiotic D-cycloserine, and following overexpression of the periplasmic protease CtpA [161,178]. Altogether, these findings strongly suggest that σ^SbrI^ plays a role in the cell-envelope stress response and directs the transition from motile cells to biofilm growth [35].

### 4.2. σ Factors Regulating the Production and Secretion of Extracellular Enzymes, Toxins, and Toxic Secondary Metabolites

#### σ^VreI^

The σ^VreI^ factor controls the expression of Hxc-T2SS and the PdtA/PdtB two-partner secretion (TPS) system, among other genes [179]. These secretion systems mediate the secretion of low-molecular-weight alkaline phosphatases LapA, LapB, and LapC [180] and the high-molecular-weight surface-exposed protein PdtA [152], respectively. Importantly, overproduction of this σ factor was found to enhance *P. aeruginosa* virulence in a zebrafish embryo model of infection [179]. σ^VreI^ is expressed under inorganic phosphate (Pi)-starvation conditions together with the VreR anti-σ factor in a PhoB-dependent manner [152]. PhoB also promotes the transcription of σ^VreI^ regulon genes by binding to DNA and enhancing the binding of σ^VreI^-loaded RNAP [152]. Antibodies against proteins of the σ^VreI^ regulon are found in the serum of CF patients, suggesting that this σ factor is active during infection [179]. This is supported by the fact that σ^VreI^ is active during *P. aeruginosa* infection of zebrafish embryos and lung epithelial cells [155]. Importantly, the lack of *vreI* was found to significantly reduce virulence of *P. aeruginosa* in zebrafish embryos and its cytotoxicity towards lung epithelial cells [155].

### 4.3. σ Factors Regulating the Acquisition and Homeostasis of Iron

#### 4.3.1. σ^PvdS^ and σ^FpvI^

The σ^PvdS^ factor promotes the transcription of the pyoverdine biosynthetic locus, the main siderophore produced by *P. aeruginosa*. Furthermore, it was found that there is a correlation between genes required for biofilm formation and for pyoverdine production and that biofilm formation may be necessary for full pyoverdine production [181]. Transcriptomic analyses in presence of phenazines and oxidative stress showed that pyoverdine signaling may also be connected to these other aspects of *P. aeruginosa* pathogenesis and stress responses [119,182]. In addition to the pyoverdine biosynthetic locus, the *P. aeruginosa* σ^PvdS^ regulon comprises about 80 genes [51]. σ^PvdS^ activates the transcription of several extracellular virulence factors such as exotoxin A, PIV/PrpL protease, the T3SS effectors ExoT and ExoS, and the T3SS major regulator ExsA [51,183]. Inhibiting the expression of σ^PvdS^ was found to suppress *P. aeruginosa* pathogenicity in a mouse model of lung infection, evidencing the link between σ^PvdS^ and virulence [184].

Activity of σ^PvdS^ is controlled by the FpvR anti-σ factor and the FpvA outer membrane receptor in response to the presence of pyoverdine [185]. This signaling cascade also controls the activation of the σ^FpvI^ factor, which promotes the expression of the *fpvA* gene and several pyoverdine biosynthesis genes, the *hasR* heme receptor gene, and the *prrF1* sRNA gene [169].

#### 4.3.2. σ^HasI^ and σ^HxuI^

*P. aeruginosa* possesses two IS σ^ECF^ factors that respond to the host iron carrier heme, i.e., σ^HasI^ and σ^HxuI^ [81]. While σ^HxuI^ is activated by free heme, σ^HasI^ is activated by the heme–HasAp hemophore complex [81]. Several genes in the Has and Hxu systems are highly expressed in vivo in murine models of acute and chronic infection [186,187]. In line with this, the lack of the heme receptor HasR was found to reduce *P. aeruginosa* growth in a mouse model of acute infection [186]. The σ^HxuI^ regulon includes several virulence genes such as *lasA* elastase, *lecB* lectin, and the phenazine biosynthetic cluster [32]. Importantly, the Hxu system has shown to be particularly active during bloodstream infections in *P. aeruginosa* clinical isolates, and its ability to cause sepsis is greatly reduced in *hxuI* mutants, while it is enhanced in σ^HxuI^ overproducing strains [61]. The lack of σ^HxuI^ was also found to reduce *P. aeruginosa* acute infection in mice, and overexpression of σ^HxuI^ was found to promote long-term infection in a murine model [32].

## 5. Concluding Remarks

The emergence and spread of antibiotic resistance among pathogenic bacteria is one of the greatest health threats worldwide. The ability of *P. aeruginosa* to resist multiple antibiotics has ranked it among the critical priority pathogens for the development of new therapeutic approaches [61]. Pathogenesis in *P. aeruginosa* originates from the multi-layered combination of more than 30 virulence factors coordinated to infect and colonize virtually any human tissue (summarized in Table 1). Success in this infectious process depends on the timely production of these virulence determinants in response to the host environment and the proper use of this battery of resources to evade the host’s defenses. This is achieved by a tightly knit network of transcriptional regulators orchestrating gene expression across the different stages of the infectious process. Disrupting the signaling cascades that coordinate the expression of virulence genes has arisen as a promising alternative strategy to treat *P. aeruginosa* infections, as tampering with most of these transcriptional regulators results in significant drops in virulence. Many of the new therapies being developed against this pathogen are based on targeting such virulence factors and their regulators. Along these lines, several compounds have been discovered to act as QS inhibitors by blocking AHL sensing and/or production or as structural analogues of PQS that inhibit quinolone signaling in *P. aeruginosa* [44]. Disrupting iron homeostasis is also flourishing as a new therapy to treat *P. aeruginosa* infections with the use of iron mimics and chelators, and iron acquisition mechanisms are being used to deliver antibiotic compounds with the so-called ‘Trojan-horse’ strategy [61]. Drugs targeting other virulence hallmarks of *P. aeruginosa* such as biofilm formation or its secretion systems and effectors are also under development [188]. Thus, understanding the signaling pathways controlling virulence is critical to the discovery of novel drugs against this pathogen based on the inhibition or activation of its transcriptional regulators.

## Figures and Tables

**Figure 1 ijms-24-11895-f001:**
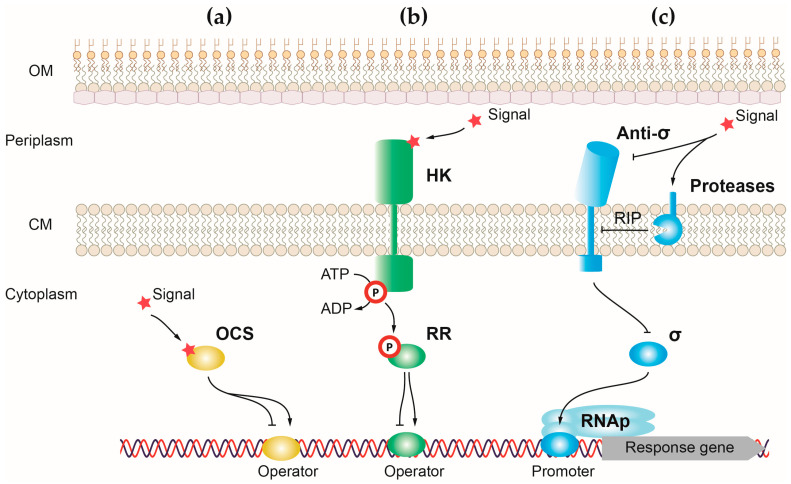
Schematic representation showing the three main classes of transcriptional regulators in *P. aeruginosa*. Mechanisms underlying gene expression regulation by (**a**) one-component systems, (**b**) two-component systems, and (**c**) alternative σ factors. Positive and negative controls are represented with arrows and T-shaped lines, respectively. OM, outer membrane; CM, cytoplasmic membrane; RIP, regulated intramembrane proteolysis.

**Figure 2 ijms-24-11895-f002:**
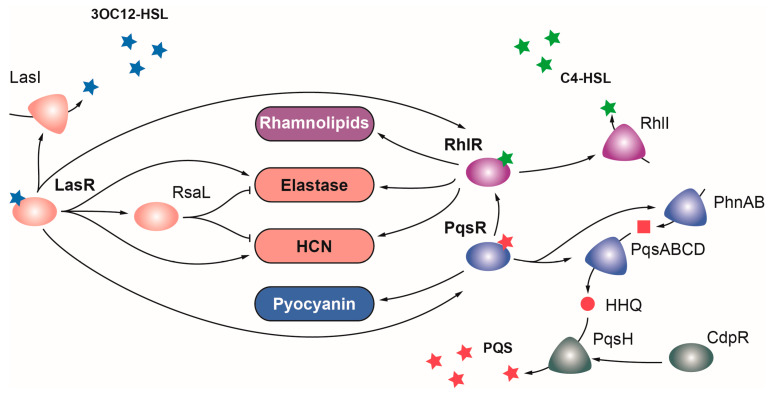
Regulation of virulence by quorum sensing in *P. aeruginosa*. The Las (depicted in pink), Rhl (depicted in purple), and PQS (depicted in blue) QS systems are governed by the OCS LasR, RhlR, and PqsR, respectively. Upon detection of their signal molecules, each OCS activates the expression of virulence factors and enzymes producing their respective inducing signals. Positive and negative regulations are represented with arrows and T-shaped lines, respectively.

**Figure 3 ijms-24-11895-f003:**
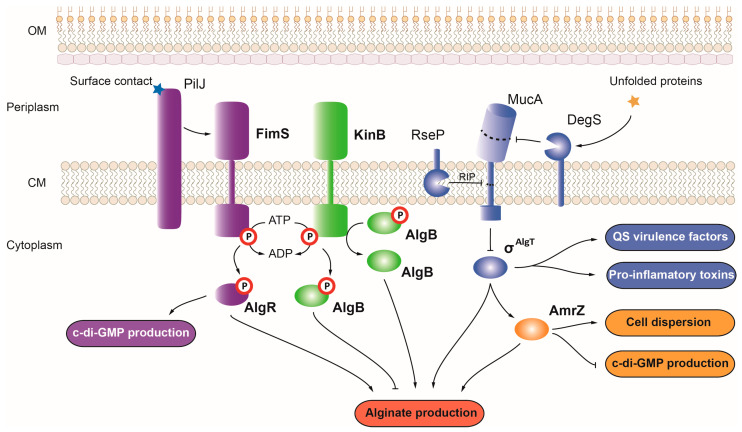
Regulation of alginate production and virulence in *P. aeruginosa*. Alginate production in *P. aeruginosa* is controlled by the OCS AmrZ (depicted in yellow), the TCSs FimS-AlgR and KinB-AlgB (depicted in purple and green, respectively), and the σ^AlgT^ factor (depicted in blue). Positive and negative controls are represented with arrows and T-shaped lines, respectively. OM, outer membrane; CM, cytoplasmic membrane; RIP, regulated intramembrane proteolysis.

**Figure 4 ijms-24-11895-f004:**
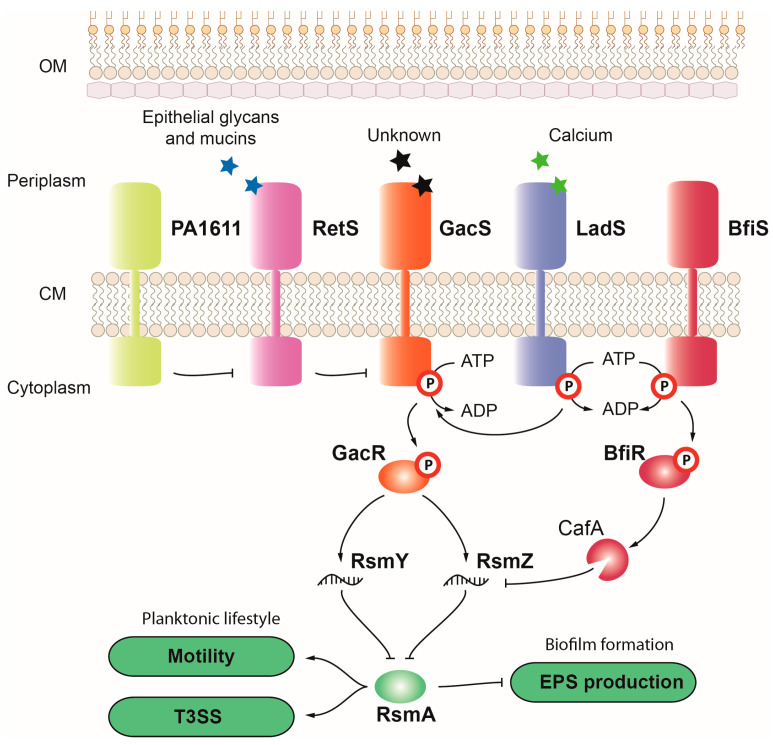
TCSs governing the switch between a planktonic lifestyle and biofilm formation via RsmA. The global regulator RsmA promotes a planktonic lifestyle by inducing motility and the production of T3SS in *P. aeruginosa* while blocking EPS production and biofilm formation. RsmA is antagonized by the two sRNAs RsmY and RsmZ. The TCSs GacS-GacR and BfiS-BfiR control RsmA activity by modulating RsmY and RsmZ levels. Positive and negative controls are represented with arrows and T-shaped lines, respectively. OM, outer membrane; CM, cytoplasmic membrane.

## Data Availability

Not applicable.

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
