# Peer review of "Transcriptional Regulators Controlling Virulence in Pseudomonas aeruginosa"

_ijms, 2023, doi:10.3390/ijms241511895_

Round 1

Reviewer 1 Report

See attached PDF file. 

The quality of the English is pretty good, only requiring minor edits. 

Author Response

Please, see attached file.

Reviewer 2 Report

In this article by Sánchez-Jiménez et al. (2023), the authors reviewed in a very complete way the main virulence factors expressed by Pseudomonas aeruginosa. The work presents tables and figures that complement the understanding of the subject in a clear and objective way. Some minor points need to be answered before publication of this manuscript, please see below.

1) lines 28-30: please include a reference supporting this epidemiological data.

2 line 67: does this topic address virulence factors of P. aeruginosa in monotype biofilms? Please make this information clear throughout the manuscript.

 3) What about biofilms formed by more than one species? Are they the same virulence factors? Is there this information in the literature? If not, I suggest pointing it out as a perspective for future studies.

4) Are all cellular and molecular signaling pathways authored by the authors? Or adapted from other papers?

Author Response

Please, see attached file
